# Meroterpenoids and Steroids from the Marine-Derived Fungus *Trametes* sp. ZYX-Z-16

**DOI:** 10.3390/molecules27248782

**Published:** 2022-12-11

**Authors:** Ziming Ren, Li Yang, Qingyun Ma, Qingyi Xie, Haofu Dai, Kunlai Sun, Youxing Zhao

**Affiliations:** 1Zhejiang Provincial Engineering Technology Research Center of Marine Biomedical Products, School of Food and Pharmacy, Zhejiang Ocean University, Zhoushan 316022, China; 2Haikou Key Laboratory for Research and Utilization of Tropical Natural Products, Institute of Tropical Bioscience and Biotechnology, Chinese Academy of Tropical Agricultural Sciences & Hainan Key Laboratory for Protection and Utilization of Tropical Bioresources, Hainan Academy of Tropical Agricultural Resource, Haikou 571101, China

**Keywords:** marine-derived fungi, *Trametes* sp., spiromeroterpenoid, α-glucosidase inhibitory activity, antibacterial activity

## Abstract

Marine fungi can metabolize structurally diverse active components, and have become an important source of drug lead molecules. In the present study, the chemical investigation on the EtOAc extract of the fermentation broth of the marine-derived fungus *Trametes* sp. ZYX-Z-16 led to the isolation of eight meroterpenoids (**1**–**8**), including two undescribed ones, together with ten ergostane steroid analogues (**9**–**18**). The structures of two new spiromeroterpenoids, asnovolin H (**1**) and asnovolin I (**2**), were determined based on 1D, 2D NMR, and HRESIMS spectroscopic data along with ECD spectra calculations. All compounds were tested for antibacterial and α-glucosidase inhibitory activity. Among them, compound **12** showed definite antibacterial activities against *Staphylococcus aureus* ATCC 6538 (MIC 32 μg/mL) and *Bacillus subtilis* ATCC 6633 (MIC 16 μg/mL). In addition, compounds **9** and **10** showed superior inhibitory activity, with IC_50_ values of 104.1 and 111.3 μM, respectively, to the positive control acarbose (304.6 μM).

## 1. Introduction

The ocean has a unique and complex ecological environment with low temperature, high temperature, high pressure, high salinity, no light, and low oxygen. The rapidly increasing number of marine natural products (MNPs) with structural novelty and high biological activities each year has greatly stimulated the interest of researchers [1,2]. In recent years, the metabolites of marine microorganisms have become one of the research hotspots of scholars at home and abroad [3]. Marine fungi form different metabolic pathways and adaptation mechanisms in the peculiar marine environment and can produce natural secondary metabolites with unique chemical structures [4]. A great deal of work on mining the active secondary metabolites from marine-derived fungi has been performed. The main types of these metabolites contained alkaloids [5,6], anthraquinones [7], terpenes [8,9,10], polyketones [11], macrolides [12], isocoumarins [13], and sterols [14]. These different types of compounds show diverse biological activities, including antibacterial [15], antifungal [16], antioxidant [17], cytotoxic [8,18], anti-inflammatory [19], enzyme inhibition [20], antituberculosis [21], neuroprotection [22], and antiproliferative [23] activities.

The genus *Trametes* of fungi is distinguished by a white aerial mycelia. In the present day, researchers have mainly focused on the biodegradable enzymes [24,25] and polysaccharides with cytotoxic activity [26,27] in the macrofungi species of genus *Trametes* such as *T. versicolor* and *T. robiniophila*. The active secondary metabolites from the microfungi species of genus *Trametes* relatively lack studies. In the course of our ongoing search for structurally new and biologically active metabolites from marine-derived fungi, the fungus *Trametes* sp. ZYX-Z-16, isolated from conch snails in Silver Island, Xisha, South Sea, China, and presenting complex metabolite profiles revealed by high-pressure liquid chromatogram, attracted our attention. Subsequent chemical investigation on the EtOAc extract of the fermentation broth of this fungus led to the identification of eight meroterpenoids (**1***–***8**), including two new spiromeroterpenoids, asnovolin H (**1**) and asnovolin I (**2**), together with ten ergostane steroid analogues (**9***–***18**) (Figure 1). Herein, the isolation, structural elucidation, and bioassays, including antibacterial, antifungal, and *α*-glucosidase inhibitory activities, of these compounds are described.

## 2. Results and Discussion

### 2.1. Structural Elucidation of Compounds

Compound **1** was isolated as a yellow powder, and the molecular formula was determined as C_26_H_38_O_7_ on the basis of HRESIMS ion peak at *m/z* [M + Na]^+^ 485.2528 (calcd. 485.2510 for C_26_H_38_NaO_7_) and ^13^C NMR data, suggesting eight degrees of unsaturation. The IR spectrum revealed the presence of hydroxyl group (3475 cm^−1^), double bond (1637 cm^−1^), ester carbonyls (1686 cm^−1^), and ketone carbonyl (1735 cm^−1^). The ^1^H NMR spectral data (Table 1) of **1** revealed the signals for seven methyls (*δ*_H_ 1.03, 1.14, 1.25, 1.40, 1.50, 1.52, and 1.72), one methoxy (*δ*_H_ 3.74), and oxygen-bearing methine (*δ*_H_ 3.95). The ^13^C NMR and DEPT (Table 1) spectra presented 26 carbon signals for eight methyls, four methylenes, five methines (one oxygenated at *δ*_C_ 69.6), and nine nonprotonated carbons (two olefinic at *δ*_C_ 183.4 and 110.2, two oxygenated, one ketone at *δ*_C_ 195.8, and two ester carbonyls at *δ*_C_ 177.9, 173.3). Three carbonyls and one double bond account for four degrees of unsaturation, indicating that four rings (A–D) are presented in compound **1**. Careful comparison of the above-mentioned NMR data of **1** with those of asnovolin A [28] revealed that they had the same spiromeroterpenoid skeleton. The only difference between them was that the methylene at C-7 (*δ*_C_ 24.4) in asnovolin A was oxidized to hydroxylated methine (*δ*_C_ 69.6) in **1**, which was confirmed by key HMBC correlations (Figure 2) from H_3_-12 (*δ*_H_ 1.14) and H_2_-6 (*δ*_H_ 1.87/1.78) to C-7, and the sequential ^1^H–^1^H COSY cross peaks of H_2_-6/H-7/H-8. The other key HMBC and ^1^H–^1^H COSY correlations of **1,** as shown in Figure 2, further corroborated the structure of **1,** as shown in Figure 1.

The relative structure of **1** was established from the analysis of ROESY data (Figure 3) and its similar NMR data with that of asnovolin A. The ROE cross peaks of H-5/H-7/H-11*β*/H-8/H-6’, H-5/H_2_-11, H-1α/H-11α, and H_3_-12*/*H_3_-13 were observed, suggesting that H-5, H-7, H-8, H-1α, and H_2_-11 resided on the same side of the A/B ring and H_3_-12*/*H_3_-13 were located at the opposite face of the A/B ring. The ROE correlations of H-1*α*/H-11*β*/H_3_-7′/H-5′ and H-8/H-11*β*/H-6′ indicated that the two sets of protons were located at the two sides of the C/D ring, respectively (Figure 3). In order to further confirm the relative configuration of the A/B ring and the C/D ring, an NMR chemical shifts calculation for the two diastereomers, (5*R*,7*S*,8*S*,9*S*,10*S*,1′*R*,5′*S*,6′*S*)-1 and (5*R*,7*S*,8*S*,9*S*,10*S*,1′*S*,5′*R*,6′*R*)-**1,** was performed. The ^13^C and ^1^H NMR data of (5*R*,7*S*,8*S*,9*S*,10*S*,1′*R*,5′*S*,6′*S*)-**1** fitted with the experimental data better than (5*R*,7*S*,8*S*,9*S*,10*S*,1′*S*,5′*R*,6′*R*)-**1** based on the correlation coefficient (R^2^), mean absolute error (MAE), and corrected mean absolute error (CMAE) parameters (Figure 4). In addition, DP4 + analysis also predicted (5*R*,7*S*,8*S*,9*S*,10*S*,1′*R*,5′*S*,6′*S*)-**1** as the most likely structure of **1,** with 100% DP4+ probability (all data). Thus, the relative configuration was assigned as 5*R**,7*S**,8*S**,9*S**,10*S**, 1′*R**, 5′*S**,6′*S*,* as shown in Figure 1. The absolute configurations of **1** were determined by the electronic circular dichroism (ECD) calculation. As shown in Figure 4, the ECD spectra of **1** displayed one positive ECD cotton effect around 270 nm and one negative ECD cotton effect around 302 nm, which were closely similar to those of asnovolin A [28]. The comparison of experimental and calculated ECD curves (Figure 5) of **1** established its absolute configuration as 5*R*, 7*S*, 8*S*, 9*S*, 10*S*, 1′*R*, 5′*S*, 6′*S*. The structure of **1** was therefore defined as 7-hydroxy-asnovolin A and named asnovolin H.

Compound **2** was isolated as a colorless amorphous solid and possessed a molecular formula C_27_H_40_O_7_ with eight degrees of unsaturation deduced from its HRESIMS ion peak at *m/z* 499.2695 [M + Na]^+^ and ^13^C NMR data. The IR spectrum also revealed the absorptions 3486 cm^−1^ for hydroxyl group, 1736 cm^−1^ for ketone carbonyl, and 1633 cm^−1^ for double bonds. The ^1^H NMR spectrum of **2** exhibited signals indicative of nine methyl groups (*δ*_H_ 0.61,1.36, 1.37, 1.43, 1.59, 1.77, 2.06, 3.69, and 3.84), including two methoxy groups (Table 1). The ^13^C NMR and DEPT (Table 1) spectra presented 27 carbon signals for nine methyls, five methylenes, two methines, and eleven nonprotonated carbonyls (four olefinic, two oxygenated, and three carbonyls). Three carbonyl (*δ*_C_ 184.2, 174.2, and 167.6) and two double bonds (*δ*_C_ 179.6, 107.9, 155.2, and 132.4) account for five degrees of unsaturation, indicating that three rings are present in **2**. The NMR spectra of **2** resembled those of asnovolin C (**4**) [28], except for the presence of a double bond at C-5’ (*δ*_C_ 132.4) and C-6’ (*δ*_C_ 155.2) in **2** replacing two corresponding sp^3^ methine in asnovolin C. The relative configuration of **2** was determined to be the same as that of asnovolin C on the basis of their similar 1D NMR data and detailed ROESY data analysis of **2** (Figure 3). The ROE correlations of H-5/H_2_-2/H-8 are indicative of *α* orientation for these protons. The ROE correlations of H_3_-12/H-11*β*, and H-1/H-11α/H-7′ and H_3_-13 suggested that the two sets of protons were located at the different sides of the C/D ring. The ECD spectra of **2** revealed positive ECD Cotton effects around 246 nm and 298 nm, and one negative ECD Cotton effect around 220 nm and 315 nm, which fitted the calculated ECD curve (Figure 5) of **2** and established its absolute configurations as 5*R*, 8*S*, 9*S*, 10*S*, 1′*R*. Therefore, compound **2** was identified and named asnovolin I.

The structures of known compounds, including five meroterpenoids (**3**–**8**) and ten ergostane steroids (**9**–**18**), were identified as asnovolin C 5′6′-dehydro hydrogen (**3**) [28] and asnovolin C (**4**) [28], chermesin A(**5**) [29], chrodrimanin E (**6**) [30], chrodrimanin H (**7**) [30], thailandolide B (**8**) [31], ergosta-4,6,8,22*E*-tetraene-3-one (**9**) [32], 14*α*-hydroxyergosta-4,7,22*E*-triene-3,6-dione (**10**) [33], dankasterones A (**11**) [34], 4*α*-hydroxy-17-methylincisterol (**12**) [35], ergosta-6*α*-hydroxy-4,22*E*-dien-3-one (**13**) [36], fortisterol (**14**) [37], gymnasterone (**15**) [38], 15*β*-methoxy-ergosta-4,6,8(14),22*E*-tetraene-3-one (**16**) [38], ganodermaside C (**17**) [39], and ganodermaside A (**18**) [40] by comparing their NMR data to the previously reported data.

Meroterpenoids are a class of natural products derived from mixed biosynthetic origin, with a terpenoid as one of the structural components [41]. Here, the isolated meroterpenoids (**1**–**8**) from this fungus were divided into two groups. One group was spiromeroterpenoids (**1**–**5**), with a spiro carbon at (C-9) through an oxygen atom. The other group was meroterpenoids (**6**–**8**) with 6/6/6/6/6 pentacyclic ring framework, which was often found in fungal metabolites [42]. Only 11 spiromeroterpenoid analogues have been reported from fungal metabolites, including asnovolins A–G [28], chermesins A–D [29], and novofumigatonin [43]. Two undescribed spiromeroterpenoids, asnovolins H and I, from marine-derived fungus *Trametes* sp. ZYX-Z-16 would further enriched the understanding of structural diversity of this type of spiromeroterpenoids.

### 2.2. Bioassays of Compounds

#### 2.2.1. α-Glucosidase and Acetylcholinesterase Inhibitory Activity

Compounds **9** and **10** showed superior α-glucosidase inhibitory activity with IC_50_ values of 104.1 µM and 111.3 µM, respectively, to that of positive control acarbose (304.6 µM). The remaining sixteen compounds (**1**–**8** and **11**–**18**) did not show remarkable inhibitory activities (IC_50_ > 200 µM) against α-glucosidase. In addition, none of these compounds showed inhibitory activity against acetylcholinesterase (IC_50_ > 200 µM).

#### 2.2.2. Antibacterial Activity

The inhibitory activity of all compounds against four bacteria (*Staphylococcus aureus* ATCC6538, *Bacillus subtilis* ATCC 6633, *Escherichia coli* ATCC 25922, and *Listeria monocytogenes* ATCC 1911) was evaluated using the 96-well microtiter plates method. The results showed that compound **5** has weak inhibitory activity against *S. aureus* (MIC = 128 µg/mL), while it is worth noting that compound **12** has good inhibitory activity against *S. aureus* (MIC = 32 µg/mL) and B. subtilis (MIC = 16 µg/mL). The remaining sixteen compounds (**1**–**4**, **6**–**11**, and **13**–**18**) did not show remarkable antibacterial activities (MIC > 128 µg/mL) against *S. aureus* and *B. subtilis*. In this assay, none of these compounds showed inhibitory activity against *E. coli* ATCC 25922 and *L. monocytogenes* ATCC 1911 (MIC > 128 µg/mL).

#### 2.2.3. Antifungal Activity

The inhibitory activities of all compounds against five phytopathogenic fungi (*Fusarium oxysporum f. sp. cubense*, *Fusarium spp.*, *Peronophythora litchii*, *Colletotrichum gloeosporioides,* and *Hylocereus undatus*) were evaluated using the broth microdilution method. Unfortunately, none of the compounds showed definite inhibitory activity.

Meroterpenoids are the main characteristic class of natural products generated by this fungus. Those spiromeroterpenoids with a spiro carbon at (C-9) through an oxygen atom were reported to show bioactivities of suppression of fibronectin expression [28] and antibacterial activity [29]. The other type of meroterpenoids with 6/6/6/6/6 pentacyclic ring framework showed various bioactivities, including insecticidal activity [30,44], lipid droplet formation inhibition [45], protein tyrosine phosphatase 1B inhibitory activity [46], and antiviral activity [42,47]. From the above bioassays of all the isolated meroterpenoids, only compound **5** showed definite inhibitory activity against *Staphylococcus aureus* ATCC6538. None of the meroterpenoids with 6/6/6/6/6 pentacyclic ring framework had good biological activity in the present experiment. 

## 3. Materials and Methods

### 3.1. General Experimental Procedures

NMR spectra were recorded on Bruker AV-500 and Bruker AV-600 spectrometers (Bruker, Bremen, Germany) with TMS as an internal standard and the peak signals of MeOD (δ_C_/_H_ 49.0/3.31) and CDCl3 (δ_C_/_H_ 77.16/7.26) as reference. The mass spectrometric (HRESIMS) data were acquired using an API QSTAR Pulsar mass spectrometer (Bruker, Bremen, Germany) and an AB SCIEX Trip TOF 5600+ mass spectrometer (SCIEX, Framingham, MA, USA). Optical rotations were measured with a JASCO P-1020 digital polarimeter (Jasco, Tokyo, Japan). IR spectra were recorded on a Shimadzu UV2550 spectrophotometer (Shimadzu, Kyoto, Japan). UV spectra and ECD data were collected using a JASCO J-715 spectropolarimeter (Jasco, Tokyo, Japan). Semipreparative HPLC was carried out using an ODS column (YMC-pack ODS-A, 10 × 250 mm, 5 µm, 4 mL/min, YMC, Kyoto, Japan). Column chromatography was carried out on silica gel (SiO2, 200–300 mesh, Qingdao Marine Chemical Inc., Qingdao, Shandong, China), and Sephadex LH-20 (green herbs, Beijing, China) and Rp-C18 (20–45 µm; Fuji Silysia Chemical Ltd., Durham, NC, USA) were used for column chromatography.

### 3.2. Fungal Material

The fungus strain Trametes sp. ZYX-Z-16 with white mycelium was isolated from an unidentified sea snail collected from Silver Island, Xisha, South Sea, China, in May 2021. After grinding, the sample (1 g) was diluted to 10^−2^ g/mL with sterile H_2_O, 100 μL of which was spread on a PDA medium plate containing chloramphenicol as bacterial inhibitor. A single colony was identified according to its morphological characteristics and ITS4 gene sequences (GenBank accession no. ON386187, see Appendix A). A reference culture of Trametes sp. ZYX-Z-16 was deposited in our laboratory and maintained at −80 °C.

### 3.3. Fermentation, Extraction, and Isolation 

The marine fungus Trametes sp. ZYX-Z-16 was cultured in the medium which contained 20 g/L malt, 20 g/L Mannitol, 10 g/L glucose, sodium glutamate 10 g, 3 g/L yeast extract, 1 g/L corn steep liquor, 0.5 g/L KH_2_PO_4_, 0.3 g/L MgSO_4_·7H_2_O, 10 g/L sea salt, and 1 L H_2_O at pH 6.5. Fungal mycelia were cut and transferred aseptically to 1 L Erlenmeyer flasks, each adding 300 mL of sterilized liquid medium. The flasks were incubated at room temperature (about 27 °C~34 °C) for 30 days.

The whole culture broth (50 L) was harvested and filtered to yield the mycelium cake and liquid broth. The mycelium cake was extracted by tissue crusher using EtOAc for three times. The EtOAc solution was evaporated under reduced pressure. A total of 135 g EtOAc extract was obtained. The extract was extracted between petroleum ether and 90% methanol (1:1) to remove the oil. The secondary metabolites extract (38 g) was subjected to a silica gel VLC column, eluting with a stepwise gradient of petroleum ether–EtOAc (10:1, 8:1, 6:1, 4:1, 2:1, 1:1, 1:2, 0:1, *v/v*) to yield eight subfractions (Fr. 1–Fr. 8) based on HPLC and TLC.

Fraction 2 (4.2 g) was further separated into four subfractions, 2.1–2.4, by reversed phase silica gel (ODS) using stepwise gradient elution with MeOH–H_2_O (40–100%). Subfraction 2.3 (452 mg) was subjected to HPLC over ODS (MeOH/H_2_O, 80:20, *v/v*) to give compound **9** (t_R_ 48 min, 13.1 mg). Fraction 3 (0.8 g) was further separated into six subfractions, 3.1–3.6, by ODS column using stepwise gradient elution with MeOH–H_2_O (30–100%). Subfraction 3.3 (92 mg) was further subjected to HPLC purification using ODS column (MeOH/H_2_O, 90:10, *v/v*) to obtain compounds **11** (t_R_ 15 min, 4 mg), **14** (t_R_ 25 min, 4.6 mg), and **15** (t_R_ 38 min, 1.9 mg). Subfraction 3.4 (70 mg) was further subjected to semipreparative HPLC using ODS column (MeOH/H_2_O, 85:15, *v/v*) to yield compounds **10** (t_R_ 22 min, 1.9 mg) and **12** (t_R_ 30 min, 2.7 mg). Fraction 4 (2.8 g) was further purified and afford seven subfractions, 4.1–4.7, by ODS using stepwise gradient elution with MeOH–H_2_O (30–100%). Subfraction 4.4 (56 mg) was further subjected to HPLC purification using ODS column (MeCN/H_2_O, 80:20, v/v) to yield compounds **17** (t_R_ 25 min, 1.5 mg) and **18** (t_R_ 38 min, 3.5 mg). Subfraction 4.7 (486mg) was further subjected to HPLC purification using ODS column (MeOH/H_2_O, 90:10, *v/v*) to afford compounds **13** (t_R_ 36 min, 2.0 mg) and **16** (t_R_ 48 min, 2.3 mg). Fraction 5 (0.6 g) was further separated into two subfractions, 5.1–5.2, by reversed phase silica gel (ODS) using stepwise gradient elution with MeOH–H_2_O (40–100%). Subfraction 5.2 (119 mg) was subjected to HPLC over ODS (50% MeCN/H2O, *v/v*) to give compounds **6** (t_R_ 25 min, 3.0 mg) and **8** (t_R_ 28 min, 5.0 mg). Fraction 7 (2.6 g) was further separated into seven subfractions, 7.1–7.7, by reversed phase silica gel (ODS) using stepwise gradient elution with MeOH–H2O (10–100%). Subfractions 7.5 (233 mg) and 7.6 (169 mg) were further subjected to HPLC purification using ODS column (MeCN/H_2_O, 45:55, *v/v*) to yield compounds **2** (t_R_ 15 min, 2.6 mg), **3** (t_R_ 18 min, 2.1 mg), and **4** (t_R_ 25 min, 2.8 mg). Subfraction 7.7 (350 mg) was chromatographed on a silica gel column with elution of petroleum ether–EtOAc (2.7:1) to give compound 5 (2.1 mg). Compound **7** (t_R_ 43 min, 4.2 mg) was isolated from subfraction 7.3 by HPLC purification using ODS column (MeCN/H_2_O, 40:60, *v/v*). Fraction 8 (2.5 g) was further separated into three subfractions, 8.1–8.3, by reversed phase silica gel (ODS) using stepwise gradient elution with MeOH–H_2_O (10–100%). Subfraction 8.2 (74 mg) was further subjected to HPLC purification using ODS column (MeCN/H_2_O, 30:70, *v/v*) to yield compound **1** (t_R_ 27 min, 8.0 mg).

Asnovolin H (**1**): Yellow amorphous powder; [α]^25^_D_ –122.0° (c 0.1, MeOH); UV (MeOH) λ_max_ (logε) 266 (3.19), 226 (2.43), 205 (2.51) nm; IR (KBr) ν_max_ cm^−1^ 3425, 1735, 1686, 1637, 1410, 1028; ECD (MeOH) λmax (∆ε): 205 (6.65), 270 (11.91), 303 (–18.63) nm; ^1^H and ^13^C NMR data, see Table 1; HRESIMS m/z [M+Na]^+^ 485.2528 (calcd. 485.2510 for C_26_H_38_NaO_7_).

Asnovolin I (**2**): colorless, amorphous solid; [α]^25^_D_ –8.0° (c 0.1, MeOH); UV (MeOH) λ_max_ (log ε) 304 (2.51), 272 (2.14), 246 (2.77), 235 (2.74), 220 (2.83) nm; IR (KBr) ν_max_ cm^−1^ 3486, 1736, 1633, 1412, 1096; ECD (MeOH) λmax (∆ε): 197 (7.57), 220 (–7.78), 247 (3.80), 267 (1.46), 298 (7.46), 334 (–8.40) nm; ^1^H and ^13^C NMR data, see Table 1; HRESIMS m/z [M+Na]^+^ 499.2695 (calcd. 499.2666 for C_27_H_40_NaO_7_).

### 3.4. Bioassay of Enzyme Inhibition, Antibacterial, and Antifungal Activity

All the compounds were evaluated for their inhibitory effects against α-glucosidase using p-NPG as the substrate, referring to the previous method [48] with acarbose and genistein as the positive control, as well as against acetylcholinesterase using AICI as the substrate, referring to the previous method [49] with Tacrine as the positive control.

Four bacteria (Staphylococcus aureus ATCC6538, Bacillus subtilis ATCC 6633, Escherichia coli ATCC 25922, and Listeria monocytogenes ATCC 1911) were used for antibacterial evaluation for all compounds, referring to the previous method [50] with ampicillin as a positive control. The bacteria inhibitory activity of all compounds was observed after 12 h in a constant temperature incubator at 37 °C. Five phytopathogens (*Fusarium oxysporum* f. sp. *cubense*, *Fusarium* spp., *Peronophythora litchii*, *Colletotrichum gloeosporioides*, and *Hylocereus undatus*) were used for antifungal evaluation for all compounds, referring to the previous broth microdilution method [51] with carbendazim as a positive control. The fungal inhibitory activity of all compounds was observed after 48 h in a constant temperature incubator at 28 °C. The MIC value was defined as the lowest concentration of the test compound at which the microorganism did not demonstrate visible growth. Each test was performed in triplicate.

### 3.5. Computation Section

Conformational search was performed using the iMTD-GC method imbedded in Crest program [52]. Density functional theory calculations were performed using the Gaussian 16 package [53]. The conformers within 5 kcal/mol were optimized at B3LYP/6-31G(d) in gas phase and the conformers with population over 1% were kept. Then, these conformers were further reoptimized at B3LYP/6-311G(d) with IEFPCM solvent model, and frequency analyses of all optimized conformers were also performed at the same level of theory to exclude imaginary frequencies. NMR shielding tensors were calculated with the GIAO method [54] at mPW1PW91/6-31+G(d,p) level with IEFPCM solvent model in methanol. The shielding constants were converted into chemical shifts by referencing to TMS at 0 ppm (δcal = σTMS – σcal), where the σTMS (the shielding constant of TMS) was calculated at the same level. For each candidate, the parameters a and b of the linear regression δcal = aδexp + b; the correlation coefficient, R^2^; the mean absolute error (MAE), defined as Σn |δcal – δexp|/n; and the corrected mean absolute error, CMAE, defined as Σn |δcorr–δexp|/n, where δcorr = (δcal – b)/a, were calculated. DP4+ probability analysis was performed using the shielding tensors [55]. ECD spectra were calculated by the TDDFT methodology at the B3LYP/TZVP, utilizing IEFPCM in methanol. ECD spectra were simulated using SpecDis 1.71 [56].

## 4. Conclusions

The metabolites from marine fungi have extensive biological and pharmacological activities and have been an important source of drug lead molecules. The chemical investigation of the marine fungus *Trametes* sp. ZYX-Z-16 led to the isolation of eight meroterpenoids (**1***–***8**), including two undescribed spiro-ones, asnovolins H (**1**) and I (**2**), together with eight ergostane steroids (**9***–***18**). The bioassay of enzyme inhibition assay showed that two ergostane steroids (**9** and **10**) revealed definite inhibition against *α*-glucosidase with IC_50_ values of 104.1 μM and 111.3 μM, respectively. The result of antibacterial assay showed that an ergostane steroid (**12**) has obvious inhibitory effect on *Staphylococcus aureus* ATCC6538 (MIC 32 μg/mL) and *Bacillus subtilis* ATCC6633 (16 μg/mL). This study further deepens the understanding of the structural diversity of meroterpenoids of marine fungi, enriches the marine natural product database, and provides theoretical information for the subsequent utilization and development of marine natural product resources.

## Figures and Tables

**Figure 1 molecules-27-08782-f001:**
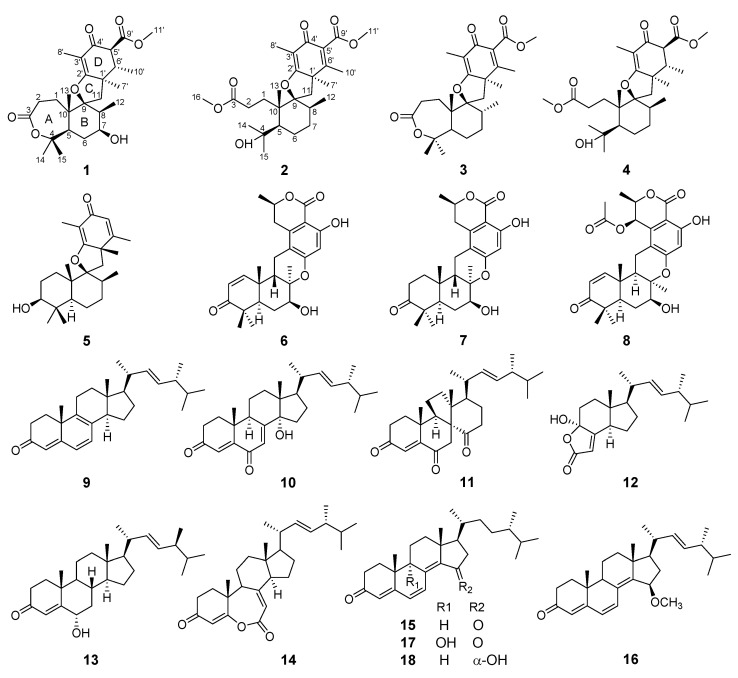
The structures of compounds **1–18**.

**Figure 2 molecules-27-08782-f002:**
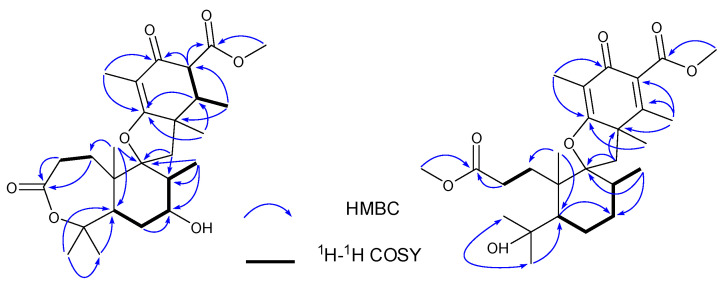
Key HMBC (H→C) and COSY (↔) correlations of **1** and **2**.

**Figure 3 molecules-27-08782-f003:**
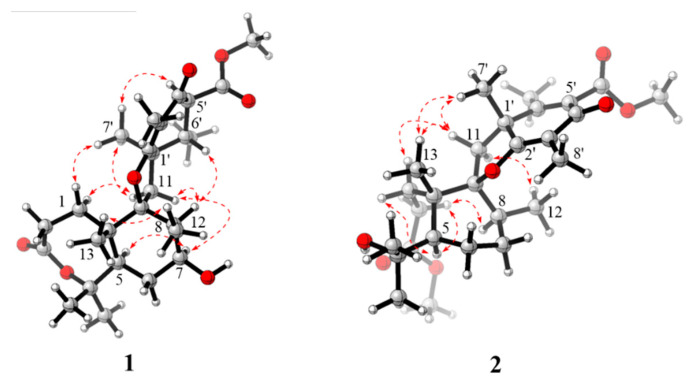
Key ROESY (red dashed arrows) correlations of compounds **1** and **2**.

**Figure 4 molecules-27-08782-f004:**
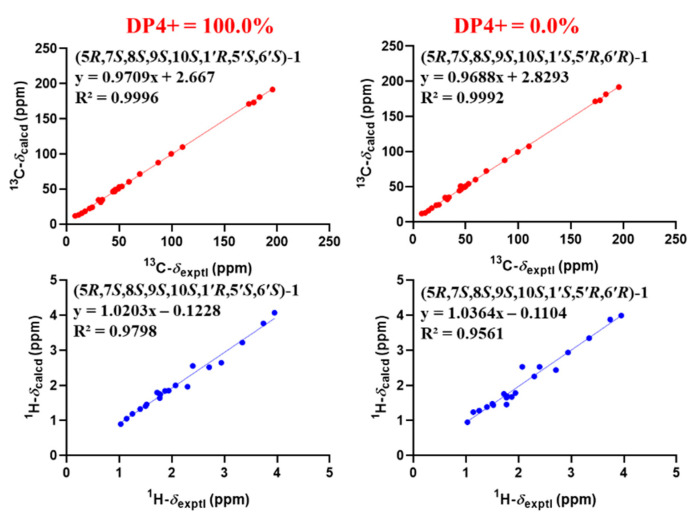
Linear regression analysis between experimental and calculated ^13^C and ^1^H NMR chemical shifts of diastereomers of (5*R*,7*S*,8*S*,9*S*,10*S*,1′*R*,5′*S*,6′*S*)-**1** (**left**) and (5*R*,7*S*,8*S*,9*S*,10*S*,1′*S*,5′*R*,6′*R*)-**1** (**right**).

**Figure 5 molecules-27-08782-f005:**
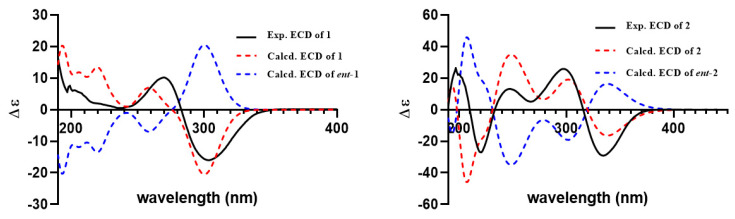
The experimental and calculated ECD spectra for **1** and **2** (σ = 0.30 eV, UV shift = 0 nm).

**Table 1 molecules-27-08782-t001:** ^1^H (500 MHz) and ^13^C NMR (125 MHz) data of compounds **1** (in MeOD) and **2** (in CDCl_3_).

No.	1		2	
	*δ* _C_	*δ*_H_ (*J* in Hz)	*δ* _C_	*δ*_H_ (*J* in Hz)
1	32.6, CH_2_	α 1.77 m	34.4, CH_2_	*α* 2.25 m
		*β* 1.87		*β* 1.48 overlap
2	32.4, CH_2_	*β* 2.94 dd (16.5, 11.7)	29.1, CH_2_	2.35 t (8.2)
		*α* 2.71 dd (16.5, 7.8)		
3	177.9, C		174.2, C	
4	87.2, C		75.0, C	
5	49.3, CH	1.80 m	48.1, CH	1.57 overlap
6	32.6, CH_2_	1.77 m	23.0, CH_2_	*α* 2.01 m
				*β* 1.74 m
7	69.6, CH	3.95 m	28.8, CH_2_	α 1.48 overlap
				*β* 1.90 overlap
8	50.1, CH	2.07 m	39.2, CH	1.90 overlap
9	99.5, C		99.8, C	
10	45.0, C		45.0, C	
11	45.9, CH_2_	*α* 2.30 d (13.9)	37.1, CH_2_	α 2.18 d (13.4)
		*β* 1.94 d (13.9)		*β* 2.07 d (13.4)
12	11.6, CH_3_	1.14 d (7.3)	16.3, CH_3_	0.61 d (5.8)
13	17.7, CH_3_	1.40 s	22.8, CH_3_	1.36 s
14	34.1, CH_3_	1.50 s	32.4, CH_3_	1.37 s
15	24.5, CH_3_	1.52 s	34.8, CH_3_	1.43 s
16			52.0, CH_3_	3.69 s
1′	46.4, C		48.1, C	
2′	183.4, C		179.6, C	
3′	110.2, C		107.9, C	
4′	195.8, C		184.2, C	
5′	59.4, CH	3.34 m	132.4, C	
6′	44.1, CH	2.40 dq (13.4, 6.7)	155.2, C	
7′	22.0, CH_3_	1.25 s	35.0, CH_3_	1.59 s
8′	8.2, CH_3_	1.72 s	8.3, CH_3_	1.77 s
9′	173.3, C		167.6, C	
10′	14.6, CH_3_	1.03, d, (6.7)	16.9, CH_3_	2.06 s
11′	52.7, CH_3_	3.74 s	52.4, CH_3_	3.84 s

## Data Availability

Not applicable.

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
