# Peer review of "Meroterpenoids and Steroids from the Marine-Derived Fungus Trametes sp. ZYX-Z-16"

_molecules, 2022, doi:10.3390/molecules27248782_

Round 1

Reviewer 1 Report

Research of the metabolites from marine fungi are an interesting trend in research on the content of secondary metabolites in raw materials that can be used in pharmacy and medicine. In my opinion, the work fits the message of the Journal – Molecules.

It is difficult for me to assess the chemical part of the work.

I present my comments on the biological studies of isolated secondary metabolites from marine fungal strain Trametes sp. ZYX-Z-16:

1. In the 3.4. Bioassay of Enzyme Inhibition, Antibacterial, and Antifungal Activity subsection lines 274, 275 I propose to delete: „The MIC values were determined at the lowest concentration where test compounds inhibited fungal growth” and to add conditions of incubation.

2. The text in lines 281, 282, 283: „The MIC value was defined as the lowest concentration of the test compound at which the microorganism did not demonstrate visible growth. Each test was performed in triplicate” after all, it concerns research of activity for bacteria and fungi.

3. Detailed results of biological research (Bioassay of Enzyme Inhibition, Antibacterial, and Antifungal Activity) are not presented, I propose to add tables as supplementary materials.

Sincerely,

MS

Author Response

#Reviewer 1

Research of the metabolites from marine fungi are an interesting trend in research on the content of secondary metabolites in raw materials that can be used in pharmacy and medicine. In my opinion, the work fits the message of the Journal – Molecules. It is difficult for me to assess the chemical part of the work. I present my comments on the biological studies of isolated secondary metabolites from marine fungal strain Trametes sp. ZYX-Z-16:

  1. In the 3.4. Bioassay of Enzyme Inhibition, Antibacterial, and Antifungal Activity subsection lines 274, 275 I propose to delete: “The MIC values were determined at the lowest concentration where test compounds inhibited fungal growth” and to add conditions of incubation.

Response: Thanks for your suggestion. We delete the sentence “The MIC values were determined at the lowest concentration where test compounds inhibited fungal growth” and add conditions of incubation in the 3.4. Bioassay of Enzyme Inhibition of the revised manuscript.

  1. The text in lines 281, 282, 283: „The MIC value was defined as the lowest concentration of the test compound at which the microorganism did not demonstrate visible growth. Each test was performed in triplicate” after all, it concerns research of activity for bacteria and fungi.

Response: Thank you for your suggestion. We have reorganized and simplified this section 3.4.

  1. Detailed results of biological research (Bioassay of Enzyme Inhibition, Antibacterial, and Antifungal Activity) are not presented, I propose to add tables as supplementary materials.

Response: Thanks for your suggestion. Detailed results of biological research were added in the supplementary materials in tabular form. 

Author Response

#Reviewer 2

This manuscript described the isolation, structural elucidation, and biological activities of two new and 16 known compounds from the marine-derived fungus Trametes sp. Although the meroterpenoids 18 were rare in natural products, the elucidations for new compounds 1 and 2, especially the relative configurations, are ambiguous.

  1. The accuracy of high resolution mass is typically within 3 ppm; however, those of 1 and 2 are 3.7 and 5.8 ppm, respectively. The large errors make them doubtful.

Response: Thanks for your reminding. Despite the errors of HRESIMS of 1 and 2 were greater than 3 ppm, the HRESIMS of 1 and 2 combined with the 1D NMR spectra could provide solid evidence to determine the molecular formulas of 1 and 2 as deduced in the manuscript.

2.The NMR data at C-1, C-2, C-6 in compound 1 seem to be incorrectly assigned, as compared with data in reference 28. Practically, the protons in beta position (H2-1) of carboxylate are not likely to appear at lower field than those in alpha position (H2-2).

Response: Thanks for your reminding. The NMR data at C-1, C-2, C-6 in compound 1 were corrected and marked in red in the revised manuscript.

  1. As the configurations of B, C, and D ring in this structural type are diversified (Figure 1), the elucidation for relative configuration needs to be carefully performed. I think the relative configuration of 1 is not well elucidated in the text. The NOE correlation of H-5/H-11α (2.30) could not conclude that the proton (2.30 ppm) of C-11 is alpha-oriented. In the molecule model of 1, the distances between H-5 and H-11α and between H-5 and H-11β are less than 3 Å. This implied that both protons of C-11 might have NOEs with H-5 and in the present data the orientations of both protons at C-11 are indistinguishable. Accordingly, the present NOE data in Figure 3 could not establish the stereorelation between AB and D rings. I strongly suggested that a detail analysis of NOE correlations and another experiment, such as quantum chemical calculations, are necessary for clarifying the structure of 1. The same concern for compound 2 also needs to be clarify.

Response: We are grateful for your suggestion. A detailed ROE correlations analysis was performed for compounds 1 and 2. Furthermore, NMR chemical shifts calculation, as you required, was conducted to confirm the relative configuration of A/B ring and C/D ring as established by ROE correlations analysis.

  1. Also, the manuscript needs an extensive English editing. Part of the errors and concerns are listed as follows:

Line 41: “good” è diverse

Response: Thanks for your suggestion. We corrected the “good” as “diverse”.

Line 44, delete “are reported up to date.” and replaced by “activities”

Response: We are grateful for your suggestion. “are reported up to date.” was replaced by “activities” in the revised manuscript.

Line 69: “one proton relate to oxygenated carbon”è “oxygen-bearing methine”

Line

Response: We corrected the “one proton relate to oxygenated carbon” as “oxygen-bearing methine”.

108-109 “…1736 cm-1 for ketone carbonyl and 1633 cm-1 for double bonds and ester carbonyls….”: 1633 cm-1 is not likely to be ester carbonyl.

Response: We apologize for our error and delete the “and ester carbonyls” in the revised manuscript.

Line123: “the key sequential ROE cross peaks”: This is an inappropriate expression for NOE or ROE correlations.

Response: Thanks for your reminding. We delete the word “sequential” in the revised manuscript.

Line 185: The chemical shifts of solvent residues should be provided in the experimental section.

Response: Thanks for your reminding. The chemical shifts of solvent residues were added to the revised manuscript.

The manuscript is not well written, and the structures are ambiguously elucidated. I don’t think that it is suitable for publication in Molecules.

Response: We apologize for the confusion generated by our previous version of manuscript. We revised and polished this manuscript as you required and hope meet your requirement and suitable for publication in Molecules.

Reviewer 3 Report

Compound 1 it is 7-hydroxy-asnovolin A. 

The authors identified two new compounds, which are not fundamentally new. These compounds are simple derivatives of known compounds - Asnovolines. This information is low novelty, and the editor should decide how important chemical novelty is in this journal.

The originality of the presented information is not high, however, information about two new metabolites belonging to the Asnovolin series may be useful in the future as fundamental. Аt the same time, biological activities data are not a significant contribution.

Thus, the article is well written, and probably can be published. However, it will be cited only if another group of authors isolates compounds 1 and 2 from another object and will be forced to refer to this work. From the point of view of references to the results of biological activity, here rather will not be referenced.

So, in my opinion, the editor should decide to accept or reject this paper. 

Author Response

#Reviewer 3

Compound 1 it is 7-hydroxy-asnovolin A.

Response: Thanks for your reminding. We added the “7-hydroxy-asnovolin A” to compound 1 in the revised manuscript.

The authors identified two new compounds, which are not fundamentally new. These compounds are simple derivatives of known compounds - Asnovolines. This information is low novelty, and the editor should decide how important chemical novelty is in this journal.

The originality of the presented information is not high, however, information about two new metabolites belonging to the Asnovolin series may be useful in the future as fundamental. Аt the same time, biological activities data are not a significant contribution.

Thus, the article is well written, and probably can be published. However, it will be cited only if another group of authors isolates compounds 1 and 2 from another object and will be forced to refer to this work. From the point of view of references to the results of biological activity, here rather will not be referenced.

So, in my opinion, the editor should decide to accept or reject this paper.

Response: We are grateful for your assessment. We revised and polished this manuscript as the reviewers required and hope the revised manuscript suitable for publication in Molecules.

Round 2

Author Response

    1. The NMR calculation did not follow the instruction recommended by DP4+ method (J. Org. Chem. 2015, 80, 12526-12534). For example, the solvent mode in GIAO NMR calculation did not properly follow the suggestions, and the improper use of this method could not obtain meaningful results (Nat. Prod. Rep. 2022, 39, 58).

    Response: we are grateful for your suggestions. In our manuscript, the relative configurations of 1 and 2 could be clearly determined by the ROE correlations analysis as shown in Figure 3 and the NMR calculation was only used to further corroborate the result. Moreover, the NMR calculation we performed in the manuscript followed the instruction recommended by DP4+ method (J. Org. Chem. 2015, 80, 12526-12534). The DFT functional mPW1PW91coupled with the 6-31+G** basis set was the best among the 24 levels of theory under study (page 12 in J. Org. Chem. 2015, 80, 12526-12534) and recommended by Ariel M. Sarotti (section 4.4 (d) in Nat. Prod. Rep. 2022, 39, 58). As for the solvent model, we apologize for our typo error. We, actually, used methanol not chloroform in the solvent mode of NMR calculation. The “chloroform” was corrected as “methanol” in the computational section of the revised manuscript

    1. Moreover, the Cartesian coordinates of conformers used in NMR calculations along with their Boltzmann distributions and the raw data of DP4+ probability should be provided in SI.

    Response: Thanks for your reminding. The Boltzmann distributions of optimized conformers used in NMR calculations and the raw data of DP4+ probability were provided in the second version of SI. The cartesian coordinates of conformers used in NMR calculations were added in the third version of SI (the latest version).

    1. Most importantly, the mass data remains unmodified. Such a large error in high resolution mass data is not acceptable for a high quality scientific journal. Thus, I don’t suggest its publication in Molecules.

    Response: We apologize for the large error of the HRESIMS data, which possibly derived from the calibration error of the instrument. Unfortunately, owing to the limited amount of samples and that all the samples were consumed in the bioactivity assays, we do not have the sample to re-determine the HRESIMS of 1 and 2. However, the spectroscopic data including UV, IR, NMR, and HRESIMS and comparison of these data with those of known compounds could provide robust evidence to unambiguously determine the structures of 1 and 2 d in our manuscript.